

# The extent to which genetics and lean grade affect fatty acid profiles and volatile compounds in organic pork

Immaculada Argemí-Armengol[1], Daniel Villalba[1], Marc Tor[1],
Cristina Pérez-Santaescolástica[2], Laura Purriños[2], José Manuel Lorenzo[2] and
Javier Álvarez-Rodríguez[1]

[1] Departament de Ciència Animal, Universitat de Lleida, Lleida, Spain
[2] Centro Tecnolóxico da Carne de Galicia, Ourense, Spain

## ABSTRACT

Niche production is intended to produce premium pork, but several husbandry factors may affect the meat fatty acid composition and aroma. Fatty acid profile (by GC-FID) of raw meat and volatile compounds (by SPME-GC–MS) of cooked meat were analysed in loin samples from two pig genetic types-75% Duroc (Du) and 50% Pietrain (Pi) rossbreds that were slaughtered at different weights (90 kg and 105 kg, respectively) to achieve similar target carcass fatness, and the outcome carcasses were balanced for lean grade groups (<60% or ≥60% lean) within genotypes. Genetic type did not affect fatty acids (FA) profile of meat. The leaner meat had lower C12:0 and C20:$3n-3$, lower saturated fatty acids (SFA) and higher MUFA/SFA ratio content than the fattier meat. Short-chain alcohols were lower in Pietrain and in leaner pork compared to the samples from Duroc crossbreds and fattier pork. A greater amount of hexane,2,4,4-trimethyl (an aliphatic hydrocarbon) but lower carbon disulphide (sulphur compound) content was detected in pork from leaner compared to fattier pork. Higher aromatics hydrocarbons were exclusively associated with Duroc crossbreds, and lower aliphatic hydrocarbons with pigs classified as fattier. Most of the volatile compounds detected in the present study came from lipid oxidation.

# INTRODUCTION

The flavor of cooked meat is one of the most important sensory attributes for consumers to judge the quality and it is influenced by volatile compounds (VC), formed during cooking, contributing to the sense of taste that determine the meat aroma attributes (*Calkins & Hodgen, 2007*; *Zhao et al., 2017*).

The amount and nature of aroma precursors present in pork depends on several factors including feed, ageing, gender, *post-mortem* treatment and genetic variations (*Meinert et al., 2007*). Crossbreeding with Duroc breed as sire line allows improving meat quality traits (*Ramírez & Cava, 2007*). Likewise, Duroc breed is used to produce dry-cured pork products, since accumulates greater intramuscular fat and fat quality traits than other sire breeds as Pietrain, which is normally very lean (*Affentranger et al., 1996*; *Latorre et al.,*

Corresponding author
Immaculada Argemí-Armengol,
immaargemi@gmail.com

*2009b*). Moreover, the amount of intramuscular fat in organic pork has been reported to be higher (*Sundrum et al., 2000*), and the fatty acid composition to be more unsaturated (*Hansen et al., 2006*).

This experiment had the hypothesis that different pig genotypes, under organic husbandry, react differently to same feeding, thereby showing different fatty acid profiles in Duroc than Pietrain crossbred pork. Moreover, the arisen volatile compounds contents of this cooked pork may also be affected.

Therefore, the objective of the present study was to evaluate the potential role of genetic type and lean grade on fatty acid composition and volatile compounds of pork under organic husbandry.

## MATERIALS & METHODS

### Animal management and meat sampling

A total of 48 pigs raised by 12 sows from two genetic types were used in this study. Twenty-six animals were Pietrain × (Landrace × Large White) (Pi × (LD × LW)) and 22 animals were Duroc × (Gascon × Duroc) (Du × (Gc × Du)). Half of the animals were females; the other half were castrates. The Pi genetic types derived from Pi line of Selección Batallé (Riudarenes, Girona, Spain) and the Du genetic types derived from Du line of German Genetic (Stuttgart-Plieningen, Germany). Pigs from both genotypes were housed together in three concrete floor pens with a space allowance $\geq 2.3$ m$^2$/pig. The individual body-weight (BW) was determined at the start of the growing phase (initial age 68 $\pm$ 15 days) and the BW at slaughter was set at approximately 105 kg (Pi) or 90 kg (Du) to reach a similar carcass fatness (lean content) between genetic types (around 60% lean). Pigs were kept in accordance with the European Community standards for organic livestock and livestock products (EC-regulation 889/2008 supplementing the EC-regulation 834/2007). All pigs had the same feed ad libitum, whose ingredients were barley grain (34.8%), maize grain (20.0%), wheat grain (13.4%), expeller soybean meal (12.3%), pea (10.0%), maize germ meal (4.0%), vegetable protein concentrate (2.0%), soybean oil (0.5%), minerals (2.6%) and vitamin-micromineral premix (0.4%). The feed had 140 g of crude protein and 37 g of crude fat per kg, with an analyzed fatty acid composition of 20.96 g SFA/100 g of identified FA, 19.98 g MUFA/100 g FA, 53.24 g PUFA $n-6$/100 g FA and 5.93 g PUFA $n-3$/100 g FA, and a calculated metabolizable energy content was 12.7 MJ/kg of feed. Pigs were stunned by $CO_2$ (concentration 87%) before slaughtering (Escorxador Frigorífic d'Avinyó S.A., Catalonia, Spain) using a dip lift system, exsanguinated, scalded, skinned, eviscerated according to standard commercial procedures and split down the midline. Hot carcass weight was individually recorded before the carcasses were refrigerated in line processing at 2 °C. The carcasses were graded with an automated image analysis system (VCS 2000, E + V Technology GmbH, Oranienburg, Germany; (*DOUE, 2018*) and classified in two classes (<60%, $n = 10$ and >60% lean, $n = 14$). Backfat thickness was measured at 3rd–4th last thoracic rib. At 45 min post-mortem, the loins were excised from the carcass following the standard procedures of the abattoir and they were trimmed by an expert staff to eliminate part of the external fat for commercial requirements. Immediately

afterwards, individual 10-cm caudal *Longissimus lumborum* was sampled (approximately 500 g), packaged in polyethylene bags and stored at 4 °C in darkness overnight.

## Sample selection and preparation

One-day post-mortem, 24 *L. lumborum* pork samples were selected and two slices (1 cm thickness, ∼100 g each) were vacuum-packaged in plastic bags. The first slice was used to determinate total intramuscular fat content (IMF) and fatty acid composition of raw pork, thus it was stored at −20 °C, freeze-dried and minced until analysis. The second slice was used to determinate volatile compounds of cooked pork aged 8 days. Thereby, loin slice was aged in dark at 4 °C for one week and kept at −20 °C. Before cooking, these aged loin samples were thawed at 4 °C during 24 h, and subsequently cooked by placing the vacuum bags in a water bath (95 °C) with automatic temperature control to internal temperature of 70 °C, controlled by thermocouples connected to a data logger. After cooking, samples were cooled at room temperature overnight, vacuum-packaged and stored at −20 °C for no longer than four weeks until analysis.

## Fat and fatty acids (FA) content of meat

Loin slices were trimmed of intermuscular and subcutaneous fat prior to IMF analysis. Fat content was quantified using the Ankom procedure *AOCS (2005)* (Official Procedure Am 5–04) with an Ankom extractor (XT10; Ankom Technology, Madrid, Spain). Analyses were run in duplicate. Meat fatty acid (FA) methyl esters were directly obtained by transesterification using a solution of methanol/sulphuric acid 2% (v/v), 30 min heating at 80 °C, centrifugation at 3,000 rpm during 5 min and collection of the final supernatant. Analysis of FA methyl esters were performed in duplicate by GC with a 30 m × 0.25 mm capillary column (Agilent DB-23; Agilent Technologies, Santa Clara, CA, USA) and a flame ionization detector with helium as the carrier gas at two mL/min. The oven temperature program increased from 150 °C at the first min, to 180 °C at 35 °C per min, and to 220 °C at 5 °C per min. The injector and detector temperatures were both 250 °C. Fatty acid composition was calculated as the relative percentage of each individual FA relative to total FA. Individual FA were identified by comparing their retention times with those from a known standard Supelco® 37 Component FAME Mix (Supelco, Bellefonte, PA, USA). The proportion of polyunsaturated (PUFA) (C18: $2n-6$; C18: $3n-3$; C18: $3n-6$; C20: $2n-6$; C20: $3n-6$; C20: $3n-3$ and C20: $4n-6$), monounsaturated (MUFA) (C16: $1n-7$; C17: $1n-7$; C18: $1n-9$; C18: $1n-7c$; C20: $1n-9$) and saturated (SFA) (C10:0; C12:0; C14:0; C16:0; C17:0; C18:0; and C20:0) FA contents were calculated.

## Volatile compounds analysis (VC)

The extraction of the volatile compounds was performed using solid-phase microextraction (SPME) from the previously homogenized samples described in section 'sample selection and preparation' (pork aged 8 days). An SPME device (Supelco, Bellefonte, PA, USA) containing a fused-silica fibre (10 mm length) coated with a 50/30 mm thickness of DVB/CAR/PDMS (divinylbenzene/carboxen/polydimethysiloxane) was used and analysis was performed as following. For headspace SPME (HS-SPME) extraction, one g of each sample was weighed in a 20 mL glass vial, after being ground using a commercial grinder.

The vials were subsequently screw-capped with a laminated Teflon-rubber disc. The fibre was previously conditioned by heating in a Fiber Conditioning Station at 270 °C for 30 min. The conditioning, extraction and injection of the samples was carried out with an autosampler PAL-RTC 120. The extractions were carried out at 37 °C for 30 min, after equilibration of the samples for 15 min at the temperature used for extraction, ensuring homogeneous temperature for sample and headspace. Once sampling was finished, the fibre was transferred to the injection port the gas chromatograph-mass spectrometer (GC-MS) system.

A gas chromatograph 7890B (Agilent Technologies) was used with a DB-624 capillary column (30 m, 0.25 mm i.d., 1.4 μm film thickness; J&W Scientific, Folsom, CA, USA) coupled to a 5977B single quadrupole mass selective detector (Agilent Technologies, Palo Alto, CA, USA). The SPME fibre was desorbed and maintained in the injection port at 260 °C during 8 min. The sample was injected in split less mode. Helium was used as carrier gas with a flow of 1. two mL/min (9.59 psi). The temperature program was isothermal for 10 min at 40 °C, raised to 200 °C at 5 °C/min, and then raised to 250 °C at 20 °C/min, and held for 5 min: total run time 49.5 min. Injector and detector temperatures were both set at 260 °C.

The mass spectra were obtained using a mass selective detector working in electronic impact at 70 eV, with a multiplier voltage of 850 V and collecting data at 6.34 scans/s over the range m/z 40–550. Compounds were identified by comparing their mass spectra with those contained in the NIST14 (National Institute of Standards and Technology, Gaithersburg) library, and/or by comparing their mass spectra and retention time with authentic standards (Supelco, Bellefonte, PA, USA), and/or by calculation of retention index relative to a series of standard alkanes (C5-C14) (for calculating Linear Retention Index, Supelco 44585-U, Bellefonte, PA, USA) and matching them data reported in literature. The results are expressed as quantifier area units (AU) $\times 10^4$/g of sample.

Among the volatile compounds (VC) at the end of final stage, only compounds regarded as mainly representative for their presence and abundance to aroma have been take into account according to previous studies (*Gorbatov & Lyaskovskaya Yu, 1980*; *Flores et al., 1997*; *Machiels et al., 2003*; *Domínguez et al., 2014a*; *Franco, Vazquez & Lorenzo, 2014*; *Gravador et al., 2015*; *Benet et al., 2015*; *Zhao et al., 2017*; *Pérez-Santaescolástica et al., 2018*; *Flores, 2018*).

The VC were classified based on their origin (lipolysis, proteolysis and microbial), according to *Dainty, Edwards & Hibbard (1985)*, *Roger, Degas & Gripon (1988)*, *Ruiz et al. (1999)*, *Meynier et al. (1999)*, *Carrapiso et al. (2002)*, *Arnoldi (2003)*, *Liu (2003)*, *Machiels et al. (2003)*, *Raes et al. (2003)*, *Martín et al. (2006)*, *Ramírez & Cava (2007)*, *Calkins & Hodgen (2007)*, *Narváez-Rivas, Gallardo & León-Camacho (2012)*, *Fonseca et al. (2015)*, *Rivas, Gallardo & Camacho (2016)* and *Pérez-Santaescolástica et al. (2019)*.

## Statistical analysis

The data were analyzed with the JMP Pro 13 statistical software (SAS Institute, Cary, NC, USA) with a least square mean standard model including genetic type, lean grade and sex as fixed effects. Differences ($P \leq 0.05$) between least square means were assessed with the

Tukey test. Tendencies were reported when the *P*-value ranged between 0.05 and 0.10. A Spearman's rank correlation analysis between FA and VC content was performed.

Classification trees (partition option from multivariate methods) from JMP Pro software were used to predict both genetic type and carcass grading as a function of potential predictor variables (19 fatty acids and 69 volatiles compounds) using recursive partitioning. The partition algorithm searched all possible splits of predictors to best predict the response (FA or VC). These splits (or partitions) of the data were done recursively to form a tree of decision rules. The variables were selected according to G2 (likelihood-ratio chi-square) test of association and logworth ($-\log(p\text{-value})$) value. The logworth values are the logs of adjusted *p*-values for the chi-square test of independence. These are adjusted to account for the number of ways that splits can occur. The partition algorithm imputes -that is, randomly assigns- values for the missing values, and this allows the variables that are poorly populated to be noticed, if they indeed help explain banding. If the logworth is greater than 2, the variable that is used in the branch is considered significant and should be included in the tree.

## RESULTS

### Fatty acids composition of raw pork

The results concerning IMF and FA composition of raw pork (24 *h post-mortem*) according to genetic type and lean content is detailed in Table 1. Total IMF was not significantly affected by genetic type and lean grade ($P > 0.05$). Nineteen FA were detected and quantified, which the percentage of oleic acid ($C18:1n-9$) was the highest, ranging from 36.5% to 38.8%, followed by palmitic acid (C16:0) from 21.9% to 23% and linoleic acid ($C18:2n-6$) from 15.1% to 16.4%. No significant differences were observed in FA contents between genetic types ($P > 0.05$). However, Du pigs tended to show higher SFA content than Pi pigs ($P = 0.10$), due to a tendency for higher proportions of palmitic acid (C16:0) and myristic acid (C14:0) ($P = 0.08$). Likewise, Du pigs tended to show higher dihomo-$\gamma$-linolenic acid ($C20:3n-6$) than Pi-sired pigs ($P = 0.10$).

On the contrary, individual FA differed between lean grades, especially the SFA lauric acid (C12:0) and the omega-3 PUFA eicosatrienoic acid ($C20:3n-3$), with leaner carcasses ($\geq 60\%$ lean) showing greater lauric acid content ($P = 0.05$) and lower eicosatrienoic acid content ($P = 0.05$) than fatter carcasses ($<60\%$ lean). The leaner carcasses tended to show higher total SFA content than fatter carcasses ($P = 0.10$). Accordingly, the MUFA/SFA ratio of leaner carcasses ($\geq 60\%$ lean) was lower ($P = 0.05$) than in fatter carcasses ($<60\%$ lean). However, the PUFA/SFA ratio in raw pork was not affected by lean content ($P > 0.10$). There were significant differences ($P < 0.05$) between genders in the margaric acid (C17:0), oleic acid ($C18:1n-9$), cis-gadoleic acid ($C20:1n-9$) and MUFA/SFA ratio. Pork from females had more margaric acid (C17:0) than males ($P < 0.05$). However, the MUFA oleic acid ($C18:1n-9$) and cis-gadoleic acid ($C20:1n-9$) were higher in pork from males than in that of females ($P < 0.05$). Accordingly, the MUFA/SFA ratio was higher in pork from males than in that of females ($P < 0.05$).

**Table 1  Fatty acid (FA) composition (g/100 g identified FA) in *L. lumborum* of pork as affected by genetic type and lean grade.**

| | Genetic type | | Lean grade | | SEM | P-value[†] | |
|---|---|---|---|---|---|---|---|
| | Pi × (L × LW) | Du × (Gc × Du) | <60% lean | >60% lean | | Genetic type | Lean grade |
| N meat samples | 8 | 16 | 15 | 9 | | | |
| Fat (IMF) | 2.05 | 2.26 | 2.23 | 2.08 | 0.03 | 0.63 | 0.75 |
| Saturated FA (SFA) | | | | | | | |
| C10:0, decanoic | 0.10 | 0.07 | 0.10 | 0.07 | 0.04 | 0.56 | 0.70 |
| C12:0, dodecanoic | 0.11 | 0.10 | 0.12 | 0.08 | 0.01 | 0.64 | 0.05 |
| C14:0, tetradecanoic | 1.39 | 1.52 | 1.50 | 1.41 | 0.05 | 0.10 | 0.29 |
| C16:0, hexadecanoic | 21.91 | 22.97 | 22.55 | 22.33 | 0.39 | 0.08 | 0.73 |
| C17:0[a], heptadecanoic | 0.30 | 0.40 | 0.36 | 0.34 | 0.04 | 0.13 | 0.79 |
| C18:0, octadecanoic | 11.96 | 11.77 | 12.24 | 11.48 | 0.29 | 0.67 | 0.12 |
| C20:0, eicosanoic | 0.09 | 0.11 | 0.10 | 0.10 | 0.01 | 0.50 | 0.67 |
| Sum of SFA | 35.86 | 36.93 | 36.97 | 35.81 | 0.42 | 0.10 | 0.10 |
| Monounsaturated FA (MUFA) | | | | | | | |
| C16:1, palmitoleic | 2.58 | 2.88 | 2.64 | 2.82 | 0.26 | 0.43 | 0.68 |
| C17:1, heptadecenoic | 0.26 | 0.31 | 0.28 | 0.30 | 0.03 | 0.25 | 0.67 |
| cis-9-18:1[b], cis-9-octadecenoic | 37.51 | 37.76 | 36.51 | 38.76 | 0.91 | 0.85 | 0.14 |
| cis-11-18:1, cis-11- octadecenoic | 2.95 | 3.06 | 2.91 | 3.10 | 0.21 | 0.73 | 0.57 |
| cis-11-20:1[b], cis-9-eicosenoic | 0.57 | 0.56 | 0.57 | 0.56 | 0.02 | 0.58 | 0.85 |
| Sum of MUFA | 43.87 | 44.58 | 42.91 | 45.54 | 1.23 | 0.70 | 0.20 |
| Polyunsaturated FA (PUFA) | | | | | | | |
| C18:2 n-6, linoleic | 16.36 | 15.18 | 16.39 | 15.16 | 1.20 | 0.51 | 0.53 |
| C18:3 n-6, γ-linolènic | 0.11 | 0.10 | 0.10 | 0.11 | 0.02 | 0.74 | 0.73 |
| C18:3 n-3, α-linolenic | 1.28 | 1.19 | 1.31 | 1.16 | 0.12 | 0.62 | 0.43 |
| C20:2, eicosadienoic | 0.58 | 0.46 | 0.53 | 0.48 | 0.04 | 0.13 | 0.44 |
| C20:3 n-6, dihomo-γ-linolenic | 0.25 | 0.19 | 0.22 | 0.23 | 0.02 | 0.10 | 0.82 |
| C20:3 n-3 eicosatrienoic | 0.15 | 0.14 | 0.18 | 0.11 | 0.02 | 0.55 | 0.05 |
| C20:4 n-6, arachidonic | 1.51 | 1.19 | 1.35 | 1.36 | 0.15 | 0.16 | 0.91 |
| Sum of PUFA | 20.23 | 18.46 | 20.08 | 18.61 | 1.40 | 0.40 | 0.52 |
| PUFA/SFA ratio | 0.57 | 0.50 | 0.55 | 0.52 | 0.04 | 0.31 | 0.70 |
| MUFA/SFA ratio[b] | 1.22 | 1.21 | 1.16 | 1.27 | 0.03 | 0.77 | 0.05 |

**Notes.**

[a] Pork from gilts showed greater C17:0 than that of barrows (0.43 *vs.* 0.27 ± 0.04%).

[b] Pork from barrows showed greater *cis*-9-18:1, *cis*-11-20:1 and MUFA/SFA ratio than pork from gilts (39.4 6 v. 35.82 ± 1.04%; 0.61 v. 0.52 ± 0.02%; 1.29 v. 1.14 ± 0.03%, respectively).

[†] Interaction genetic type × lean grade non-significant in any variable ($P > 0.05$).

## Volatile compounds of cooked aged pork

A total of 69 VCs were determined and they were assigned to the following chemical families: 18 hydrocarbons, 13 aldehydes, eight ketones, seven carboxylic acids, 15 alcohols, three esters and ethers, two sulphur-containing and 3 furans. Alcohols and aldehydes accounted for the highest percentage (42.1% and 22.7%, respectively, as shown in Fig. 1A). The chemical families of VC were not affected by genetic type and lean grade ($P > 0.10$). However, Du pork tended to show higher aromatic and cyclic hydrocarbons content than Pi pork ($P = 0.10$). Fifty-nine out of the 69 identified VC were classified based on their

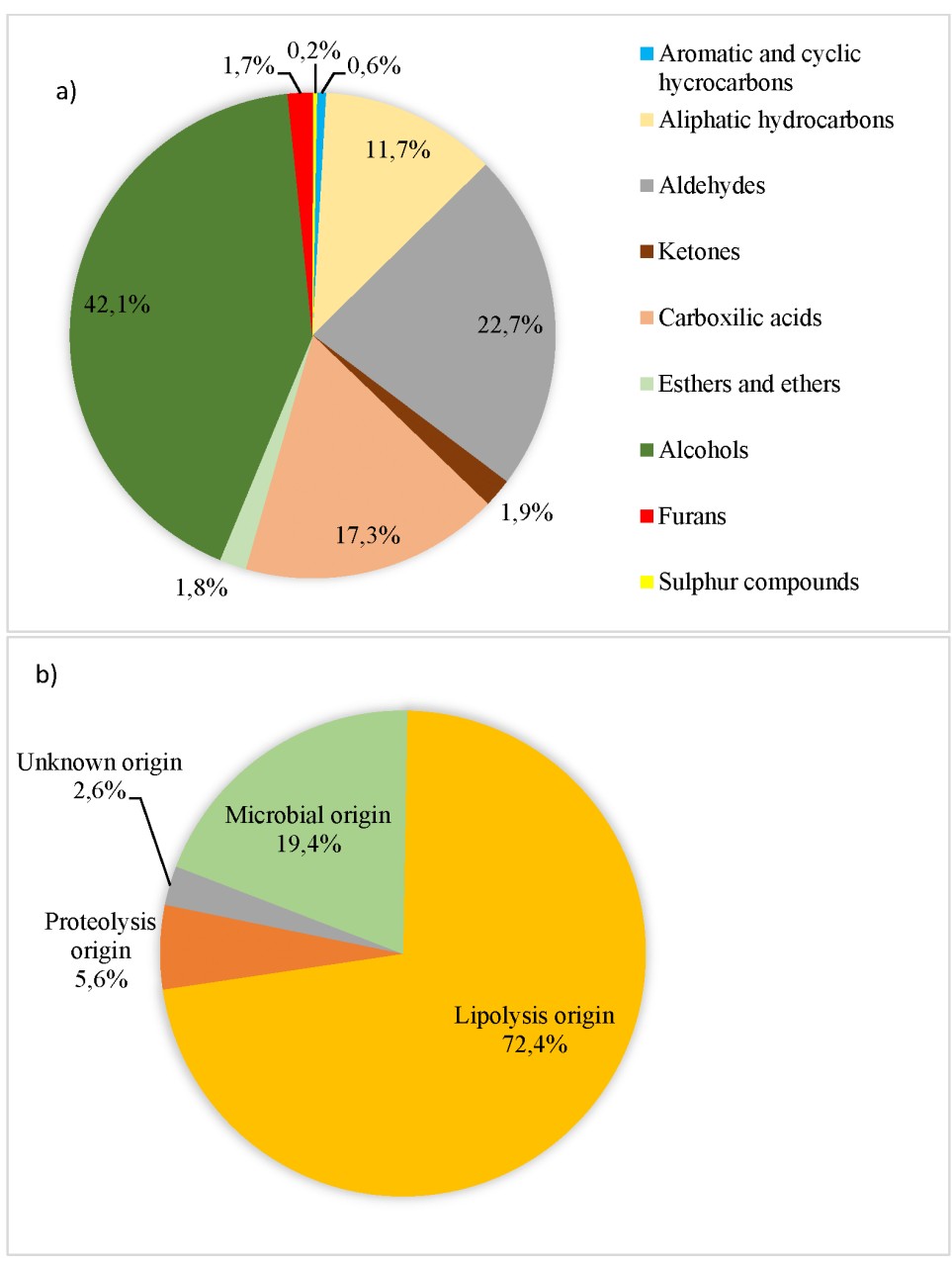

**Figure 1** Volatile compounds (AU × 104/total volatile compound AU × 104) recorded in organic pork quantified and grouped according to their chemical families (A) and their origin (B) to *Dainty, Edwards & Hibbard (1985)*, *Roger, Degas & Gripon (1988)*, *Ruiz et al. (1999)*, *Meynier et al. (1999)*, *Carrapiso et al. (2002)*, *Arnoldi (2003)*, *Liu (2003)*, *Machiels et al. (2003)*, *Raes et al. (2003)*, *Martín et al. (2006)*, *Ramírez & Cava (2007)*, *Calkins & Hodgen (2007)*, *Narváez-Rivas, Gallardo & León-Camacho (2012)*, *Fonseca et al. (2015)* and *Rivas, Gallardo & Camacho (2016)*.

origin (lipolysis, proteolysis and microbial), and 10 had an unknown origin. Lipolysis origin showed the highest level (72.4% out of total VC) (Fig. 1B), followed by proteolysis and microbial activity. The origin group of VC were not significantly influenced by the

genetic type or lean grade ($P > 0.10$). Meanwhile, pork classed as leaner had higher content of VC from unknown origin ($P < 0.01$).

The average contents of extracted VC of cooked aged pork are shown in Tables 2, 3 and 4. Through GC-MS analysis, there were several peaks in chromatogram that were not included in the list of VC, because most of them were tentatively identified as siloxanes or silanes and the most probable origin will be the trap (fiber). Concerning the effect of genetic type, there were only significant differences in the alcohol 1-pentanol content, which was greater in pork from Du than in pork from Pi pigs ($P < 0.05$). Likewise, lean grade affected the content of hydrocarbon hexane, 2,4,4-trimethyl in cooked aged pork, being greater in pigs classed as leaner ($\geq 60\%$ lean) than fattier pigs ($< 60\%$ lean) ($P < 0.05$). Nevertheless, fattier pigs ($< 60\%$ lean) had higher content of certain alcohols such as 1-butanol ($P < 0.05$) and 1-pentanol ($P < 0.05$) than leaner pigs ($\geq 60\%$ lean). The only significant differences between genders in VC were on the alcohols 1-butanol and phenylethyl, that were higher in pork from gilts than in that from barrows (6.7 vs. 2.9 $\pm$ 0.97 and 13.4 vs. 2.0 $\pm$ 3.2 AU $\times 10^4$/g of cooked pork, $P < 0.05$, respectively).

## Correlations between Volatile compounds and Fatty acids

Concerning hydrocarbons, the concentration of undecane (1-Undecene, 9-methyl-) was positively correlated ($0.41 \leq r \leq 0.53$; $P < 0.05$) with percentages of margaric acid (C17:0), heptadecenoic acid (C17: $1n-7$), linoleic acid (C18: $2n-6$), alfa-linolenic acid (C18: $3n-3$), total PUFA, and also, PUFA/SFA ratio. However, the concentration of undecane was negatively correlated ($-0.43 \leq r \leq -0.47$; $P < 0.05$) with oleic acid (C18: $1n-9$) and cis-gadoleic acid (C20: $1n-9$). The concentration of 2,4,4-trimethyl- hexane showed a negative correlation ($-0.56 \leq r \leq -0.57$; $P < 0.01$) with lauric acid (C12:0) and myristic acid (C14:0).

A positive correlation between the concentration of aldehyde 2-octanal,(E)- and myristic acid content ($r = 0.49$, $P < 0.05$), as well as between 2-octanal,(E)- and total SFA ($r = 0.55$; $P < 0.01$) was found. In contrast, the concentration of propanal, pentanal and hexanal aldehydes showed negative correlations ($-0.40 \leq r \leq -0.47$; $P < 0.05$) with contents of lauric acid (C12:0), myristic acid (C14:0) and arachidic acid (C20:0).

The concentration of 3-hydroxy-3-methyl-2-butanone (ketone) exhibited strong positive correlations ($r = 0.70$; $P < 0.001$) with contents of individual PUFA dihomo-$\gamma$-linolenic acid (C20: $3n-6$) and arachidonic acid (C20: $4n-6$). The concentration of both acetoin (ketone) and acetic acid ethenyl ester (ester) showed positive correlations ($0.62 \leq r \leq 0.65$; $P < 0.001$) with MUFA palmitoleic acid (C16: $1n-7$) content. On the contrary, the concentration of ethyl acetate, acetic acid ethenyl ester and acetoin showed negative correlations ($-0.42 \leq r \leq -0.49$; $P < 0.05$) with linoleic acid (C18: $2n-6$), gamma-linolenic acid (C18: $3n-6$), alfa-linolenic acid (C18: $3n-3$), total PUFA, and PUFA/SFA ratio.

The concentration of butanoic acid (carboxylic acid compound), 3-methyl- showed positive correlations ($r = 0.55$; $P < 0.01$) with the percentage of C14:0 and C16:0. However, the concentration of pentanoic acid had negative correlation with the content of C14:0

Argemí-Armengol et al. (2019), *PeerJ*, DOI 10.7717/peerj.7322

**Table 2  Effect of genetic type and lean grade on volatile compounds content; Hidrocarbons (expressed as AU × $10^4$/g of cooked pork).**

| Volatile compounds | LRI | R | Genetic type | | Lean grade | | SEM | P-value [†] | |
|---|---|---|---|---|---|---|---|---|---|
| | | | Pi × (L × LW) | Du × (Gc × Du) | <60% lean | >60% lean | | Genetic type | Lean grade |
| Aliphatic hydrocarbons | | | | | | | | | |
| Pentane | 517 | ms, lri,s | 81.87 | 109.60 | 96.22 | 95.24 | 36.70 | 0.62 | 0.99 |
| n-Hexane | 563 | ms, lri | 75.92 | 60.85 | 64.97 | 71.80 | 23.03 | 0.66 | 0.84 |
| Heptane | 677 | ms, lri | 25.82 | 34.41 | 37.16 | 23.07 | 14.19 | 0.69 | 0.51 |
| 1-Octene | 818 | ms, lri | 26.06 | 29.14 | 33.24 | 21.96 | 12.80 | 0.87 | 0.56 |
| Octane | 825 | ms, lri,s | 114.96 | 141.69 | 151.89 | 104.75 | 57.66 | 0.76 | 0.59 |
| Nonane | 940 | ms, lri,s | 73.77 | 158.53 | 137.17 | 95.13 | 36.03 | 0.13 | 0.44 |
| Hexane, 2,4,4-trimethyl- | 1,074 | ms, lri | 78.86 | 74.10 | 21.90 | 131.06 | 27.66 | 0.91 | 0.02 |
| Heptane, 3,3,4-trimethyl- | 1,094 | ms, lri | 3.02 | 3.34 | 3.23 | 3.13 | 0.58 | 0.72 | 0.91 |
| Undecane | 1,119 | ms, lri,s | 59.35 | 59.84 | 56.39 | 62.80 | 8.56 | 0.97 | 0.62 |
| Heptane, 4-methylene- | 1,132 | ms, lri | 11.04 | 18.94 | 18.01 | 11.97 | 4.10 | 0.21 | 0.33 |
| Dodecane | 1,194 | ms, lri,s | 36.44 | 36.73 | 34.31 | 38.86 | 5.30 | 0.97 | 0.57 |
| 1-Nonene | 1,207 | ms, lri | 1.67 | 2.84 | 1.93 | 2.58 | 1.24 | 0.53 | 0.73 |
| Tridecane | 1,264 | ms, lri,s | 14.34 | 13.95 | 13.69 | 14.60 | 2.30 | 0.91 | 0.79 |
| 1-Undecene, 9-methyl- | 1,279 | ms, | 0.65 | 0.66 | 0.53 | 0.77 | 0.10 | 0.94 | 0.11 |
| Aromatic and cyclic hydrocarbons | | | | | | | | | |
| Toluene | 807 | ms, lri | 9.27 | 9.22 | 8.65 | 9.85 | 1.77 | 0.99 | 0.65 |
| Cyclopropane, pentyl- | 819 | ms, lri | 6.26 | 33.25 | 21.94 | 17.56 | 13.40 | 0.19 | 0.83 |
| Benzene, 1,3-dimethyl- | 930 | ms, lri | 5.56 | 5.00 | 3.97 | 6.59 | 1.74 | 0.83 | 0.33 |
| Phenol, 2,6-bis(1,1-dimethylethyl) -4-(1-methylpropyl)- | 1,493 | ms, lri | 7.13 | 6.42 | 5.57 | 7.99 | 2.41 | 0.85 | 0.51 |

**Notes.**

LRI,  Lineal Retention Index calculated for DB-624 capillary column (30 m × 0.25 mm id, 1.4 μm film thickness) installed on a gas chromatograph equipped with a mass selective detector; R,  Reliability of identification;  lri,  linear retention index in agreement with literature (*Gorbatov & Lyaskovskaya Yu, 1980*; *Flores et al., 1997*; *Machiels et al., 2003*; *Domínguez et al., 2014a*; *Domínguez et al., 2014b*; *Franco, Vazquez & Lorenzo, 2014*; *Gravador et al., 2015*; *Benet et al., 2015*; *Zhao et al., 2017*; *Pérez-Santaescolástica et al., 2018*; *Flores, 2018*);  ms,  mass spectrum agreed with mass database (NIST14);  s, mass spectrum and retention time identical with an authentic standard.

[†]Interaction genetic type × lean grade non-significant in any variable ($P > 0.05$).

Argemí-Armengol et al. (2019), *PeerJ*, DOI 10.7717/peerj.7322

**Table 3** Effect of genetic type and lean grade on volatile compounds content: Aldehydes, Ketones and Carboxilic acids (expressed as AU × $10^4$/g dry matter) of cooked pork.

| Volatile compounds | LRI | R | Genetic type | | Lean grade | | SEM | P-value[†] | |
|---|---|---|---|---|---|---|---|---|---|
| | | | Pi × (L × LW) | Du × (Gc × Du) | <60% lean | >60% lean | | Genetic type | Lean grade |
| **Aldehyde** | | | | | | | | | |
| Propanal | 527 | ms, lri,s | 68.09 | 68.13 | 71.52 | 64.70 | 14.76 | 1.00 | 0.76 |
| Butanal, 3-methyl- | 661 | ms, lri | 6.68 | 18.93 | 17.97 | 7.64 | 11.79 | 0.49 | 0.56 |
| Butanal, 2-methyl- | 673 | ms, lri | 4.08 | 12.54 | 10.65 | 5.97 | 5.23 | 0.29 | 0.55 |
| Pentanal | 730 | ms, lri,s | 109.22 | 48.16 | 44.94 | 112.44 | 30.00 | 0.19 | 0.15 |
| Hexanal | 869 | ms, lri,s | 1655.35 | 845.03 | 878.54 | 1621.84 | 454.74 | 0.25 | 0.29 |
| Heptanal | 978 | ms, lri,s | 54.50 | 54.66 | 40.93 | 68.23 | 15.66 | 0.99 | 0.26 |
| 2-Heptenal, (Z)- | 1,042 | ms, lri | 4.39 | 5.42 | 4.79 | 5.01 | 0.66 | 0.31 | 0.82 |
| Benzaldehyde | 1,050 | ms, lri | 10.24 | 7.37 | 8.50 | 9.11 | 2.58 | 0.46 | 0.87 |
| 2-Butenal, (Z)- | 1,051 | ms, lri | 27.86 | 47.01 | 46.61 | 28.26 | 9.34 | 0.18 | 0.20 |
| Octanal | 1,071 | ms, lri,s | 13.35 | 14.87 | 11.18 | 17.04 | 4.22 | 0.81 | 0.36 |
| 2-Octenal, (E)- | 1,129 | ms, lri | 3.69 | 5.93 | 7.50 | 2.12 | 2.07 | 0.47 | 0.10 |
| Nonanal | 1,154 | ms, lri,s | 15.05 | 15.39 | 13.01 | 17.43 | 4.27 | 0.96 | 0.49 |
| 2,4-Decadienal, (E,E)- | 1,322 | ms, lri | 1.00 | 1.06 | 0.64 | 1.42 | 0.30 | 0.89 | 0.10 |
| **Ketone** | | | | | | | 0,00 | | |
| Acetone | 529 | ms, lri | 63.22 | 65.76 | 75.31 | 53.67 | 15.38 | 0.91 | 0.36 |
| 2-Butanone | 595 | ms, lri | 5.78 | 8.79 | 11.44 | 3.13 | 2.76 | 0.47 | 0.06 |
| 2-Pentanone | 722 | ms, lri | 5.39 | 4.09 | 6.30 | 3.18 | 1.48 | 0.56 | 0.17 |
| 2,3-Pentanedione | 738 | ms, lri | 3.74 | 2.96 | 2.48 | 4.21 | 1.00 | 0.61 | 0.26 |
| 3-Hydroxy-3-methyl -2-butanone | 820 | ms, lri | 3.71 | 0.61 | 2.54 | 1.78 | 1.17 | 0.09 | 0.67 |
| 1-Octen-3-one | 931 | ms, lri | 2.63 | 2.27 | 2.58 | 2.31 | 0.65 | 0.71 | 0.78 |
| 2-Heptanone | 971 | ms, lri | 35.59 | 42.72 | 44.93 | 33.38 | 7.94 | 0.55 | 0.34 |
| 2-Octanone | 1,064 | ms, lri | 2.68 | 2.88 | 2.07 | 3.48 | 0.74 | 0.86 | 0.22 |

Argemí-Armengol et al. (2019), *PeerJ*, DOI 10.7717/peerj.7322

Peer J

**Table 3** (*continued*)

| Volatile compounds | LRI | R | Genetic type | | Lean grade | | SEM | *P-value* [†] | |
|---|---|---|---|---|---|---|---|---|---|
| | | | Pi × (L × LW) | Du × (Gc × Du) | <60% lean | >60% lean | | Genetic type | Lean grade |
| **Carboxilic acid** | | | | | | | | | |
| Acetic acid | 697 | ms, lri | 3.66 | 5.38 | 4.77 | 4.26 | 1.91 | 0.55 | 0.86 |
| Acetoin | 790 | ms, lri | 1106.92 | 859.55 | 1330.02 | 636.45 | 579.05 | 0.78 | 0.43 |
| Butanoic acid | 923 | ms, lri | 2.17 | 1.67 | 1.37 | 2.48 | 0.45 | 0.46 | 0.11 |
| Butanoic acid, 3-methyl- | 973 | ms, lri | 5.62 | 5.05 | 8.45 | 2.22 | 4.52 | 0.93 | 0.37 |
| Pentanoic acid | 1,089 | ms, lri | 9.62 | 8.55 | 7.62 | 10.54 | 2.65 | 0.79 | 0.47 |
| Pentanoic acid, 2-methyl-, anhydride | 1,142 | ms, lri | 13.64 | 15.36 | 14.22 | 14.77 | 2.99 | 0.70 | 0.90 |
| Octanoic acid | 1,230 | ms, lri | 1.11 | 0.40 | 0.71 | 0.80 | 0.31 | 0.13 | 0.85 |

**Notes.**

LRI, Lineal Retention Index calculated for DB-624 capillary column (30 m × 0.25 mm id, 1.4 $\mu$m film thickness) installed on a gas chromatograph equipped with a mass selective detector; R, Reliability of identification; lri, linear retention index in agreement with literature (*Gorbatov & Lyaskovskaya Yu, 1980*; *Flores et al., 1997*; *Machiels et al., 2003*; *Domínguez et al., 2014a*; *Domínguez et al., 2014b*; *Franco, Vazquez & Lorenzo, 2014*; *Gravador et al., 2015*; *Benet et al., 2015*; *Zhao et al., 2017*; *Pérez-Santaescolástica et al., 2018*; *Flores, 2018*); ms, mass spectrum agreed with mass database (NIST14); s, mass spectrum and retention time identical with an authentic standard.

[†] Interaction genetic type × lean grade non-significant in any variable ($P > 0.05$).

**Table 4 Effect of genetic type and lean grade on volatile compounds content: Ester, Eter, Alcohol, Furan and Sulfur (expressed as AU × 10⁴/g dry matter) of cooked pork.**

| Volatile compounds | LRI | R | Genetic type | | Lean grade | | SEM | P-value † | |
|---|---|---|---|---|---|---|---|---|---|
| | | | Pi × (L × LW) | Du × (Gc × Du) | <60%lean | >60% lean | | Genetic type | Lean grade |
| **Ester and ether** | | | | | | | | | |
| Acetic acid ethenyl ester | 589 | ms, lri | 61.40 | 26.30 | 63.16 | 24.53 | 23.52 | 0.35 | 0.26 |
| Ethyl Acetate | 600 | ms, lri | 63.17 | 40.79 | 89.29 | 14.67 | 60.85 | 0.81 | 0.42 |
| Acetic acid, butyl ester | 1,069 | ms, lri | 13.12 | 13.86 | 9.35 | 17.63 | 3.88 | 0.90 | 0.17 |
| **Alcohol** | | | | | | | 0,00 | | |
| 1-Propanol, 2-methyl- | 649 | ms, lri | 11.40 | 31.52 | 17.28 | 25.64 | 12.84 | 0.31 | 0.67 |
| 1-Butanol | 709 | ms, lri | 3.51 | 6.10 | 6.49 | 3.12 | 0.875 | 0.06 | 0.02 |
| 3-Buten-1-ol, 3-methyl- | 805 | ms, lri | 12.20 | 23.18 | 19.89 | 15.49 | 8.98 | 0.42 | 0.75 |
| 1-Butanol, 3-methyl- | 811 | ms, lri | 138.99 | 239.02 | 215.79 | 162.22 | 81.29 | 0.42 | 0.66 |
| 1-Pentanol | 850 | ms, lri | 237.22 | 356.03 | 356.48 | 236.77 | 31.46 | 0.02 | 0.02 |
| 2,3-Butanediol | 913 | ms, lri | 61.86 | 24.44 | 57.87 | 28.43 | 19.28 | 0.20 | 0.28 |
| 2,3-Butanediol, [S-(R*, R*)]- | 921 | ms, lri | 19.05 | 18.20 | 23.89 | 13.36 | 7.98 | 0.94 | 0.34 |
| 1-Hexanol | 959 | ms, lri | 891.06 | 1886.34 | 1645.30 | 1132.10 | 413.83 | 0.12 | 0.42 |
| 1-Heptanol | 1,050 | ms, lri | 44.24 | 42.54 | 44.19 | 42.60 | 13.38 | 0.93 | 0.94 |
| 1-Octen-3-ol | 1,056 | ms, lri | 347.75 | 450.68 | 420.37 | 378.07 | 54.80 | 0.22 | 0.61 |
| 2-Ethyl-1-hexanol | 1,099 | ms, lri | 3.70 | 3.97 | 3.79 | 3.87 | 0.63 | 0.77 | 0.93 |
| 1-Octanol | 1,132 | ms, lri | 13.62 | 21.11 | 19.66 | 15.07 | 4.37 | 0.26 | 0.49 |
| Phenylethyl Alcohol | 1,189 | ms, lri | 6.03 | 9.34 | 10.72 | 4.66 | 2.93 | 0.46 | 0.18 |
| 1-Nonanol | 1,207 | ms, lri | 4.16 | 5.83 | 3.19 | 6.79 | 2.66 | 0.68 | 0.37 |
| 1-Tetradecanol | 1,472 | ms, lri | 4.04 | 4.42 | 4.76 | 3.70 | 1.68 | 0.88 | 0.67 |
| **Furans** | | | | | | | | | |
| Furan, 2-ethyl- | 706 | ms, lri | 6.67 | 10.73 | 8.77 | 8.63 | 2.09 | 0.21 | 0.97 |
| 2-n-Butyl furan | 948 | ms, lri | 4.01 | 4.54 | 4.24 | 4.31 | 0.95 | 0.71 | 0.96 |
| Furan, 2-pentyl- | 1,043 | ms, lri | 79.08 | 103.76 | 94.44 | 88.41 | 21.71 | 0.45 | 0.85 |
| **Sulphur compounds** | | | | | | | | | |
| Methanethiol | 505 | ms, lri | 3.07 | 1.43 | 3.41 | 1.09 | 1.54 | 0.48 | 0.32 |
| Carbon disulfide | 535 | ms, lri | 11.24 | 9.95 | 10.93 | 10.26 | 1.76 | 0.63 | 0.80 |

Notes.

LRI, Lineal Retention Index calculated for DB-624 capillary column (30 m × 0.25 mm id, 1.4 μm film thickness) installed on a gas chromatograph equipped with a mass selective detector; R, Reliability of identification; lri, linear retention index in agreement with literature (*Gorbatov & Lyaskovskaya Yu, 1980*; *Flores et al., 1997*; *Machiels et al., 2003*; *Domínguez et al., 2014a*; *Domínguez et al., 2014b*; *Franco, Vazquez & Lorenzo, 2014*; *Gravador et al., 2015*; *Benet et al., 2015*; *Zhao et al., 2017*; *Pérez-Santaescolástica et al., 2018*; *Flores, 2018*); ms, mass spectrum agreed with mass database (NIST14); s, mass spectrum and retention time identical with an authentic standard.

† Interaction genetic type × lean grade non-significant in any variable ($P > 0.05$).

($r = -0.43$; $P < 0.05$) but the former has a positive correlation with C18: $3n-3$ ($r = 0.41$, $P < 0.05$).

Regarding alcohols, the concentration of 1-Butanol, 3-methyl-, 2,3-Butanediol -[S-(R*, R*)]- and 1-octen-3-ol showed a positive correlation ($0.42 \leq r \leq 0.48$; $P < 0.05$) with the content of arachidonic acid (C20: $4n-6$) and dihomo-$\gamma$-linolenic acid (C20: $3n-6$). On the contrary, the concentration of 1-heptanol showed a negative correlation with the percentage of C20: $4n-6$ ($r = -0.57$; $P < 0.01$).

The concentration of carbon disulphide (sulfur compound) only showed a negative correlation with the content of lauric acid (C12:0) ($r = -0.45$; $P < 0.05$). In contrast, the concentration of furans, 2-ethyl-, 2-n-butyl, and 2-pentyl- showed a positive correlation ($0.44 \leq r \leq 0.46$; $P < 0.05$) with content of individual PUFA arachidonic acid (C20: $4n-6$).

### Partition trees

The best partition tree of FA based on genetic type resulted in two splits (Fig. 2A). The final coefficient of determination ($R^2$ square) for the validation set was 0.52. The column contributions report (based on G2) showed that PUFA/SFA ratio of raw meat and myristic acid (C14:0) are the main predictors of genetic type in the partition tree model. The lower PUFA/SFA ratio (<0.48) was exclusively associated with Du crossbred pork (100%). Within the cooked aged pork showing higher PUFA/SFA ratio ($\geq$0.48), more Pi crossbred pork (85.7%) showed lower C14:0 content (<1.4%), whereas Du crossbred pork (71.4%) had higher C14:0 ($\geq$1.4%). Another partition tree (Fig. 2B) was developed to identify VC profile of organic cooked pork based on genetic type. The final RSquare for the validation set was 0.58. The column contributions report (based on G2) showed that cyclopropane, pentyl- (cyclic hydrocarbon) and methanethiol (sulphur compound) are the main predictors of genetic type in the partition tree model. We found that higher cyclopropane, pentyl- content ($\geq$12 AU $\times$ $10^4$/g) was exclusively associated at Du crossbred (100%). Within the cooked aged pork showing lower cyclopropane, pentyl- content (<12 AU $\times$ $10^4$/g), more Du crossbred pork (66.7%) showed lower methanethiol (<0.8 AU $\times$ $10^4$/g), whereas higher methanethiol ($\geq$0.8 AU $\times$ $10^4$/g) was exclusively associated with Pi crossbred pork (100%).

The third partition tree of FA, based on lean content, showed two splits, and the proportion of observations in each split is shown in Fig. 3A. The final coefficient of determination for the validation set was 0.51. Thus, the column contributions report (based on G2) showed that total SFA content of pork and margaric acid (C17:0) are the main predictors in the partition tree model. The higher SFA content ($\geq$37.6%) was exclusively associated with pigs with higher level of fatness (<60% lean; 100%). Within the cooked aged pork having the lower SFA content (<37.5%), more fattier pigs (71.4%) showed lower C17:0 content (<0.31%), whereas leaner pigs (88.9%) had higher C17:0 ($\geq$0.31%). A fourth partition tree was developed to identify VC profile of organic cooked pork based on lean content (Fig. 3B). The final RSquare for the validation set was 0.64. The column contributions report (based on G2) showed that hexane, 2,4,4-trimethyl and carbon disulphide were the main predictors in the partition tree model. Higher hexane, 2,4,4-trimethyl content ($\geq$163 AU $\times$ $10^4$/g) was exclusively associated with pork
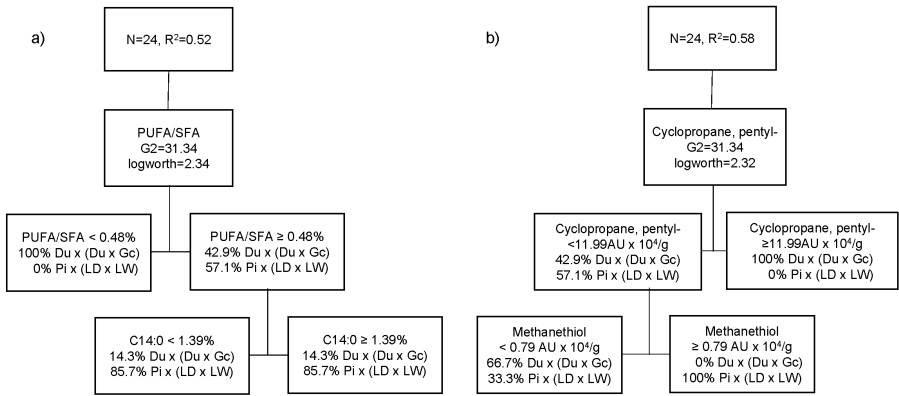

**Figure 2** Partition tree of fatty acids (A) and volatile compounds (B) of organic pork based on genetic type, showing the two splits and the proportion of observations in each split.

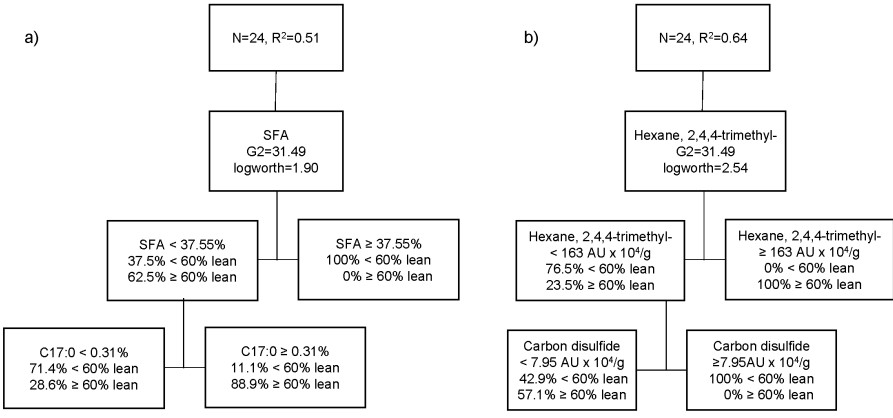

**Figure 3** Partition tree of fatty acids (A) and volatile compounds (B) of organic pork based on lean grade, showing the two splits and the proportion of observations in each split.

classified as leaner ($\geq$60% lean; 100%). Within the cooked aged pork having lower hexane, 2,4,4-trimethyl content (<163 AU $\times$ $10^4$/g), carbon disulphide content was higher ($\geq$8 AU $\times$ $10^4$/g) in pork classified as fattier (<60% lean; 100%). More pigs classified as leaner (57.1%) showed lower carbon disulphide content (<8 AU $\times$ $10^4$/g) and lower hexane, 2,4,4-trimethyl content (<163 AU $\times$ $10^4$/g) in cooked aged pork loin.

# DISCUSSION

## Fatty acids composition of raw pork

This study analyzed the relevance of genetic type and lean grade on fatty acid profiles and volatile compounds in pork under organic husbandry. These effects were assessed both independently and together (by testing its interaction). MUFA were the most prevailing group in the two pork genotypes studied (highest value for oleic acid, C18: $1n-9$), followed by SFA and PUFA. In the present study, 75%-Du crossbreds showed higher percentages

of SFA in loin compared to their 50%-Pi counterparts, which had greater MUFA/SFA ratio. However, the results indicated that the individual intramuscular FA did not differ between genetic types (Du *vs.* Pi), possibly due to targeted lack of differences in carcass and meat fatness as well as similar age between them (*Argemí-Armengol et al., 2019*). Previous research demonstrates minimal differences in fatty acid composition among similar genetic types to those used in the current study. Our results agreed with *Cameron et al. (1990)* and *Latorre et al. (2009b)*, who found only higher myristic acid (C14:0) concentrations in Du than in Pi genetics.

On the contrary, the leaner carcasses (>60% lean) had lower acid lauric (C12:0) and eicosatrienoic acid (C20: $3n-3$) but greater MUFA/SFA ratio content than the fattier meat (<60%).

While most SFA and MUFA can be efficiently synthesized *in vivo*, the lack of differences in PUFAs C18: $2n-6$ and C18: $3n-3$ proportions, which may be attributed to dietary factors (*Calkins & Hodgen, 2007*; *Juárez et al., 2017*), proves that both genetic types adapted similarly to the feed composition supply. Indeed, the SFA content of meat is positively correlated with carcass fatness (*Fiego et al., 2005*; *Latorre et al., 2009a*), which contributes to decrease the MUFA/SFA ratio and reduces the nutritional quality of pork (*Scollan et al., 2017*). Nevertheless, the lean content hardly affected the fatty acid groups either, probably because the lowest lean class was not excessively fatty, and due to their similar levels of intramuscular fat. In turn, the lower eicosatrienoic acid (C20: $3n-3$) content in leaner meat might be related to a lower feed intake pattern in lean compared to fatty pigs, which couples with their lower growth performance (*Argemí-Armengol et al., 2019*), and thereby this may reduce its biosynthesis from dietary C18: $3n-3$ (*Sibbons et al., 2018*).

## Volatile compounds of cooked aged pork

While abundant research has been conducted regarding the VC derived from pork products (cooked and/or cured with long ripening periods), less information is available for meat. Thousands of volatile compounds are generated during thermal processing that belong to various chemical classes: hydrocarbons, alcohols, aldehydes, ketones, carboxylic acids, esters, furans, sulfuric compounds and others (*Kosowska, Majcher & Fortuna, 2017*), which are produced through lipid oxidation, or through Maillard reaction or Strecker degradation (*Mottram, 1998*) or microbial degradation and others, which are responsible for meat flavour development. In the present study, higher cyclic hydrocarbons were observed in cooked loins from 75% Du compared to 50%-Pi crossbreds. More Du crossbreds were associated with greater concentrations of cyclopropane, pentyl- (an aromatic hydrocarbon), whereas more Pi crossbred pork had greater concentration of methanethiol (volatile sulphur compound from amino acid breakdown). Accordingly, methanethiol may not be a pleasant VC. Similarly, *Benet et al. (2015)* found greater methanethiol abundance in low than in high fat cooked cured ham. While most aliphatic hydrocarbons have not previously involved in meat aroma (*Flores, 2018*), methanethiol has been associated with sulfur and gasoline odor descriptors in pork broth from Chinese black-pig (*Zhao et al., 2017*) as well as with rotten eggs, sewage and cabbage in cured meat products (*Flores, 2018*). A greater amount of hexane 2,4,4-trimethyl (an aliphatic hydrocarbon) but lower carbon disulphide (sulfur compound)

was detected in pork from leaner compared to fattier pork. According to *Olivares, Navarro & Flores (2011)*, hexane is one of the volatile compounds come from lipid autooxidation. The sensory descriptions for hexane may be alkane and spicy (*Pérez-Santaescolástica et al., 2018*). *Lorenzo & Domínguez (2014)* reported that, compared with other cooking methods (grilled and roasted treatments), the application of heat for a short time led to a greater amount of hexane, and in general aliphatic hydrocarbons. In turn, carbon disulphide is an important intermediate of the Maillard reaction in the formation of heterocyclic compounds (*Mottram & Mottram, 2002*), but to our knowledge it seems uninvolved in aroma formation. Overall, even though *Sánchez-Peña et al. (2005)* reported that aliphatic hydrocarbons, because of their abundance, could play an important role in the aroma of dry-cured meat and play an important role in the overall flavor, their involvement in cooked loin primal may not be relevant.

## Correlations between Volatile compounds and Fatty acids

The concentration of short-chain alcohols (mainly 1-pentanol and butanol) were lower in Pi and in leaner pork compared to the samples from 75% Du crossbreds and fattier pork, which may counterbalance the high values of methanethiol. In this regard, 1-pentanol is produced by the degradation of homologous aldehydes during lipid and amino acid oxidation (*Garcia et al., 1991*), and it has a mild odour, fruit and balsamic aroma (*Calkins & Hodgen, 2007*), which may favour the overall aroma from Du and/or fattier pork. Most of the linear alcohols identified are oxidative decomposition products of lipids. For example, 1-butanol can come from miristoleic acid (C14:1) and 1-pentanol from linoleic acid (C18: $2n-6$), 1-hexanol may be formed from palmitoleic (C12:1) and oleic acid (C18: $1n-9$), and 1-octanol from oleic acid oxidation (*Forss, 1973*; *Flores et al., 1997*). Among the volatile compounds, the levels of 1-butanol were significantly correlated with the aromas of French dry-cured ham (*Buscailhon, Berdague & Monin, 1994*). The methyl and ethyl branched alcohols are probably derived from proteolysis, that is the Strecker degradation of amino acids (*Martín et al., 2006*; *Narváez-Rivas, Gallardo & León-Camacho, 2012*; *Fonseca et al., 2015*).

Hexane 2,4,4-trimethyl was negatively associated with lauric acid (C12:0) and myristic acid (C14:0). In addition, the most representative aldehydes (pentanal and hexanal) were also negatively correlated with C14:0. *Meynier, Genot & Gandemer (1998)* found that hexanal was the major compound from the oxidation of $n-6$ fatty acids (mainly linoleic, C18: $2n-6$ and arachidonic acid, C20: $4n-6$) in pork loin, with odor descriptions for hexanal may be green (*Zhao et al., 2017*), which oxidized rapidly when heated (*Wood et al., 2004*).

The propanal aldehyde (with sensory attribute almond-like green and toasted) was present in all samples at same level, which was the main compound family in cooked foal meat in *Domínguez et al. (2014b)*. Most PUFA (C18: $2n-6$, C18: $3n-6$, C18: $3n-3$ and C20: $2n-6$) were negatively correlated with acetoin (carboxylic acid) and acetic acid ethenyl ester (ester). The esters are formed from the interaction between free fatty acids and alcohols generated by lipid oxidation in the intramuscular tissue, specifically ethyl

esters are formed through esterification reactions between ethanol and carboxylic acids (*Peterson & Chang, 1982*).

However, arachidonic acid (C20: $4n-6$) was positively correlated with some ketones and furans, which suggest that these compounds may be derived from the oxidation of $n-6$ fatty acids, and it would prove the susceptibility of PUFA to oxidation (*Gravador et al., 2015*). In this sense, *Gandemer (2002)* found that 2-pentylfuran in meat had sensory attributes as buttery and rancid, while *Arnoldi (2003)* found that it had odour notes as fruity and sweet; which it would have been generated during heating from linoleic acid (C18: $2n-6$) oxidation (*Ruiz et al., 1999*).

## CONCLUSIONS

When pigs were slaughtered at similar carcass fatness, the 75%-Du had similar FA composition of loin meat compared to 50%-Pi genetic type, but Du genetics was prone to have low PUFA/SFA ratio (<0.48). However, the fattier carcasses (<60% lean) had a higher percentage of SFA (>37.6%) than leaner carcasses (>60% lean).

Overall, the aromatic and cyclic hydrocarbons in Du were higher than in Pi pork, which had higher content of carbon disulphide (sulphur compound). The Du and fattier pork presented higher amount of pentanol and butanol (alcohols), whereas leaner carcasses had the higher content of hexane 2,4,4-trimetil (aliphatic hydrocarbon). Most of the volatile compounds detected in the present study came from lipid oxidation.

The present results suggest that lean content rather than genetic type affected the FA composition of pork from pigs under organic husbandry which are slaughtered at light live-weights. However, the volatile compounds of cooked pork were dependent on both genetic type and lean grade. Understanding the contribution of each factor and their interactions will help the pork industry in the production of consistent premium products.

## ACKNOWLEDGEMENTS

The authors thank the owners from 'Gestió Agroecològica Porcina' farm (Solsona, Catalonia, Spain), and are indebted to 'Escorxador Frigorífic d'Avinyó' (Catalonia, Spain) for kindly supplying pork samples.

### Funding

José Manuel Lorenzo is a member of the MARCARNE network, funded by CYTED (ref. 116RT0503). The funders had no role in study design, data collection and analysis, decision to publish, or preparation of the manuscript.

### Grant Disclosures

The following grant information was disclosed by the authors:
CYTED: ref. 116RT0503.

## Competing Interests

Javier Álvarez-Rodríguez is an Academic Editor for PeerJ.

## Author Contributions

- Immaculada Argemí-Armengol, Daniel Villalba and Marc Tor performed the experiments, analyzed the data, contributed reagents/materials/analysis tools, prepared figures and/or tables, authored or reviewed drafts of the paper, approved the final draft.
- Cristina Pérez-Santaescolástica, Laura Purriños and José Manuel Lorenzo contributed reagents/materials/analysis tools, approved the final draft.
- Javier Álvarez-Rodríguez conceived and designed the experiments, performed the experiments, analyzed the data, contributed reagents/materials/analysis tools, prepared figures and/or tables, authored or reviewed drafts of the paper, approved the final draft.

## Animal Ethics

The following information was supplied relating to ethical approvals (i.e., approving body and any reference numbers):

We hereby confirm that the care and slaughter procedures of the animals were in accordance with the Spanish Policy for Animal Protection RD53/2013, which meets the European Union Directive 2010/63 on the protection of animals used for experimental and other scientific purposes.

## Data Availability

Data are available as Supplemental Files.

## Supplemental Information

Supplemental information for this article can be found online at http://dx.doi.org/10.7717/peerj.7322#supplemental-information.

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
