# Peer review of "The extent to which genetics and lean grade affect fatty acid profiles and volatile compounds in organic pork"

_PeerJ, doi:10.7717/peerj.7322_

## Round 0.1 · original submission · Minor Revisions

Dear authors,

I received now comments from reviewers concerning your and advise that your paper is acceptable after some minor revision. Please, I ask you to revise it accordingly and resubmit it as soon as possible.

With kind regards

Reviewer 1 ·

Basic reporting

The manuscript entitled "To what extend genetics and lean grade affect fatty acid profiles and volatile compounds in organic pork?" is well written, interesting, and well planned. The outcomes obtained in this study are of great value to the field of Food Science, specially for Meat quality researchers and related professionals. Tables and figures are adequate to indicate the results and the discussion is suitable and contains updated studies. I would like to recommend few improvements.

Experimental design

no comment

Validity of the findings

no comment

Additional comments

Dear authors,
The manuscript aimed to evaluate the impact of genetic and lean grade on fatty acid profile and volatile compounds in organic pork. The focus of the study is in agreement with current lines of research that connects consumers demands/desires with high quality scientific evaluation to answer relevant questions for the field. Finally, I would like to suggest some improvements.
Line 24: Please define the meaning of "Pi"
Line 26: higher proportion/content? Please check this sentence
Lines 54-55: Please check the parenthesis in this passage
Line 145: Please indicate the meaning of VC
Line 350: I would like to recommend that this sentence be update to: The estimated concentration of short-chain...
Line 361: Please remove "origin"
Line 363: ecorrelated?
Lines 386-387: Please check this sentence

Reviewer 2 ·

Basic reporting

The work is interesting inasmuch as nowadays quality products have a great relevance and can influence consumer acceptance.
The aim of this study was to evaluate the potential role of genetic type and lean grade on fatty acid composition and volatile compounds of pork under organic husbandry. I think that the topic of the manuscript is interesting for the journal and contributes to the knowledge of each factor and their interactions to improve nutritional quality of pork meat.

Experimental design

This work has a good scientific quality and is well designed.

Validity of the findings

The work needs to revise some questions to be published. I recommend a minor revision.

Additional comments

Comments to authors (e.g. suggestions of changes to the text.):
Abstract
Line 24: Please define the meaning of "Pi" and “Du”
Introduction
LINE 45. Change “… fatty acids profile …” by “… fatty acid profiles …”.
Material and methods
Line 54-55: Please check the nomenclature and use it throughout the text.
LINE 97. Insert the reference for IMF analysis.
LINE 163. What is the meaning of “spearman??”
Results
LINE 181. Change “… (P > 0.10) …” by “… (P > 0.05) …”
LINE 185. Introduce “of” before “as a consequence …”
LINES 201-224. I think is better to present first the results of the figure 1 and then the results of the Tables 2, 3 and 4.
Discussion
This section should be separated into subsections as those used in results section. This would facilitate the reading of the work.
LINES 359-360. Change this reference because it is related with dry-cured meat products. Please change it for another where the comparison is with a similar product to the one evaluated in the present work.
LINES 369-370. I understand that you mentioned regarding the abundant research has been conducted regarding the VC derived from pork products, and less information is available for meat. However, this reference should be replaced by another more appropriate.
Below, I show you some references that could be useful:
- Domínguez, R., Gómez, M., Fonseca, S., & Lorenzo, J. M. (2014). Influence of thermal treatment on formation of volatile compounds, cooking loss and lipid oxidation in foal meat. LWT-Food Science and Technology, 58(2), 439-445.
- Lorenzo, J. M., & Domínguez, R. (2014). Cooking losses, lipid oxidation and formation of volatile compounds in foal meat as affected by cooking procedure. Flavour and fragrance journal, 29(4), 240-248.
Tables
TABLE 1. Check the nomenclature of fatty acids according to “Domínguez et al. (2018). Effect of linseed supplementation and slaughter age on meat quality of grazing cross-bred Galician x Burguete foals. Journal of the Science of Food and Agriculture, 98(1), 266-273” and use it throughout the text.

Reviewer 3 ·

Basic reporting

The article represents an excellent contribution to understand how the genetics and lean grade affect fatty acid profiles, and volatile compounds of meat of the pigs reared under organic husbandry. The article is well written.

Experimental design

The experimental design is adequate. The methods are described with sufficient detail and information to replicate.

Validity of the findings

The article covers the subject with scientific accuracy. The conclusions are well stated and support the results.

Additional comments

Some detailed comments below:
Line 17 write in full FA when it appears for the first time. See all manuscript for similar issues.
Line 19 Please replace “pig genetic types (Duroc vs. Pietrain crossbreds)” with “pig genetic types- Duroc (Du) and Pietrain (Pi) crossbreds”.
Line 36 Please replace “amount and nature of aroma precursors presents” with “amount and nature of aroma precursors present”
Line 77 Please replace “measured at 3rd-4th last rib” with “measured at 3rd-4th lumbar vertebrae”
Line 108 Please check for uniformity. For example, C18:3 n-6 and in the same line C18:2n-6; C18:3n-3. See all manuscript for similar issues.
Line 114 volatile compounds with volatile compounds (VC)
Line 163 delete ??

---

## Round 0.2 · accepted · Accept

Dear authors,

I advise that your paper is now acceptable in its present form. I would like thank your for the changes that improved the quality of the manuscript.

With kind regards
Mohammed